# Neoadjuvant Treatment in Localized Pancreatic Cancer of the Elderly: A Systematic Review of the Current Literature

**DOI:** 10.3390/cancers17050747

**Published:** 2025-02-22

**Authors:** Elena Orlandi, Stefano Vecchia, Elisa Anselmi, Ilaria Toscani, Massimo Guasconi, Gennaro Perrone, Chiara Citterio, Filippo Banchini, Mario Giuffrida

**Affiliations:** 1Department of Oncology-Hematology, Azienda USL of Piacenza, 29121 Piacenza, Italy; e.orlandi@ausl.pc.it (E.O.); e.anselmi@ausl.pc.it (E.A.); i.toscani@ausl.pc.it (I.T.); c.citterio@ausl.pc.it (C.C.); 2Department of Pharmacy, Azienda USL of Piacenza, 29121 Piacenza, Italy; s.vecchia@ausl.pc.it; 3Department of Medicine and Surgery, University of Parma, 43121 Parma, Italy; massimo.guasconi@unipr.it; 4Department of Health Professions Management, Azienda USL of Piacenza, 29121 Piacenza, Italy; 5General Surgery Unit, Department of Medicine and Surgery, Parma University Hospital, 43126 Parma, Italy; gennaro.perrone82@gmail.com; 6Department of General Surgery, Azienda USL of Piacenza, 29121 Piacenza, Italy; f.banchini@ausl.pc.it

**Keywords:** pancreatic cancer, elderly patients, neoadjuvant therapy, resectable pancreatic cancer, borderline pancreatic cancer

## Abstract

Pancreatic cancer is a highly aggressive disease, and elderly patients, though they represent a numerically significant population, often face additional challenges due to comorbidities and lower tolerance to intensive treatments. Neoadjuvant therapy has been increasingly used to improve surgical outcomes and overall survival, but its role in elderly patients remains unclear. This systematic review evaluates current evidence on NAT in elderly patients with resectable, borderline, or locally advanced pancreatic cancer. We analyzed survival outcomes, resection rates, and treatment-related toxicity. Results suggest that NAT can increase R0 resection rates and improve survival in elderly patients, despite a higher incidence of adverse events. Personalized treatment regimens, such as less intensive chemotherapy protocols, may enhance tolerability without compromising efficacy. These findings support the inclusion of elderly patients in NAT protocols, emphasizing the importance of individualized treatment planning based on functional status rather than age alone.

## 1. Introduction

Pancreatic cancer is the seventh leading cause of cancer-related mortality worldwide, with higher prevalence in developed countries [1]. Despite treatment improvements, the 5-year survival rate remains around 9%, reflecting the poor prognosis associated with this disease, as indicated by a 94% mortality-to-incidence ratio [1]. The median age at diagnosis is approximately 70 years, and 68.4% of cases occur in patients over the age of 65 [2].

In early-stage resectable pancreatic tumors, the treatment of choice is curative-intent surgery, often followed by adjuvant chemotherapy [3]. Although surgery is recommended as the standard treatment for stage I and II pancreatic tumors, only about 20% of patients are eligible for surgical intervention at diagnosis due to the presence of advanced or metastatic disease [4,5]. Pancreatic resection is associated with perioperative morbidity rate of 40–60% and mortality rate of 2–3% [6]. Despite surgical treatment, prognosis remains poor, with a median overall survival ranging from 17 to 36 months [7].

Neoadjuvant therapy (NAT) gaining interest as an option to improve surgical outcomes. Neoadjuvant treatment may reduce tumor size, control micrometastatic disease, and allow clinicians to exclude patients who develop metastases during treatment, thereby potentially reducing perioperative morbidity [8]. Neoadjuvant therapy could also avoid surgery for the 15–20% of patients who develop metastatic disease; the presence of metastases precludes surgical resection [9,10]. Additionally, neoadjuvant therapy may shrink borderline tumors and sterilize their margins, increasing the likelihood of achieving R0 resection [11,12].

Although recent studies on resectable pancreatic cancer have cast doubt on the utility of neoadjuvant therapy [13] its role in borderline resectable disease appears to be more established. Evidence from randomized trials supports its efficacy in this setting [14,15,16] to the extent that neoadjuvant therapy has become the standard of care for borderline resectable pancreatic cancer [17].

NAT has become an established treatment approach in several gastro-intestinal malignancies, demonstrating improved surgical outcomes and survival rates even in elderly patients. In gastric cancer, NAT has been shown to enhance R0 resection rates and overall survival, including in the elderly patient subgroup, compared to those undergoing surgery alone [18]. Furthermore, in colorectal cancer, neoadjuvant chemotherapy is a well-integrated strategy, particularly for rectal cancer, where it has been linked to a reduction in local recurrence rates and improved disease-free survival, even in older populations [19].

While there is no unanimous consensus on the use of neoadjuvant therapy in elderly patients, limited evidence suggests that the efficacy and safety of this approach may be comparable between patients over 70 years old and younger populations [5,20]. The impact of neoadjuvant therapy in elderly patients remains underexplored; only few studies stratified patients by age, and median age in clinical trials typically falls below that of the general pancreatic cancer population [21].

In the decision-making process for the proper treatment in elderly patients, the current literature lacks robust data on whether age alone should dictate therapeutic approaches. Rather, assessing the patient’s performance status (PS) is crucial [22]. Although elderly patients are underrepresented in clinical trials, real-world studies suggest a survival benefit for elderly patients who underwent neoadjuvant chemotherapy [22]. The literature also supports that elderly patients surgically treated after NAT, despite being more susceptible to comorbidities and complications, can achieve survival outcomes comparable to younger patients [23].

To date, the literature lacks a systematic review specifically addressing the role of NAT in elderly patients.

This review aims to evaluate the short- and long-term outcomes of neoadjuvant therapy in resectable, borderline resectable (BR), and locally advanced (LA) pancreatic cancer in elderly patients, including overall survival (OS), dropout, NAT regimens analysis, time-to-recurrence (TTR), surgical resection rate, and major toxicities.

## 2. Materials and Methods

### 2.1. Information Sources

A systematic review to investigate the role of neoadjuvant therapy in resectable, borderline (BR) and locally advanced (LA) pancreatic cancer in the elderly was performed. The primary endpoint was overall survival, secondary endpoints were time-to-recurrence, surgical resection rate, and major toxicities related to NAT.

PICOS criteria were set as outlined in Table 1.

The research project was registered in the International Prospective Register of Systematic Reviews (PROSPERO) with the protocol number CRD42024503616.

An extensive bibliographic search of the literature according to the PRISMA (Preferred Reporting Items for Systematic Reviews and Meta-Analyses) 2020 checklist (Appendix A).

A systematic search of MEDLINE, EMBASE, Cochrane Database of Systematic Reviews, and Cochrane Central Register of Controlled Trials was performed. The reference lists of included articles and relevant reviews were examined to identify additional relevant publications for inclusion.

### 2.2. Inclusion and Exclusion Criteria

The present study included observational prospective and retrospective cohort studies, Randomized Controlled Trials (RCT) reporting original data, published in full-text format; studies about elderly (defined as those aged more than 70 years) patients with resectable, borderline or locally advanced pancreatic cancer reporting OS; and, if available, TTR, major toxicities, and surgical resection rate were also included.

The data analysis of the present study was performed collecting data on elderly patients defined as older than 70 years in order to reduce risk of selection bias.

Four studies used a cut-off of ≥70 years [4,6,24,25], while the remaining eight studies considered patients ≥ 75 years [26,27,28,29,30,31,32].

There were no restrictions on publication date or language. Case reports, case series, preclinical studies, and animal studies were excluded; studies involving metastatic setting were also excluded.

### 2.3. Data Extraction

A search string in MEDLINE was performed and included relevant mesh in the research field mixed with Boolean operator “AND” “OR”; the MeSH search headings used were “pancreatic”, “elderly”, and “neoadjuvant”. The other databases were searched using combinations of free text and Medical Subject Headings terms. The reference lists of the included studies were manually searched for any relevant publications.

Two reviewers (E.O. and M.G.) independently reviewed identified studies in duplicate: they screened titles and abstracts of all references identified from the initial search. The full-text articles of potentially relevant publications were scrutinized by the two reviewers in detail and inclusion criteria were applied to select eligible articles. Agreement was recorded at each stage and reported as a kappa statistic. Disagreements between reviewers were resolved through consensus.

From each eligible study, two reviewers independently extracted relevant information, using a pre-specified standardized extraction form. Data were extracted from the selected studies and entered into a standardized data collection form. The following variables were recorded: first author, year of publication, publishing journal, country, sample size, comparison group, participant demographics (age, sex, number of elderly patients, number of elderly patients who underwent NAT and/or RT, ECOG PS), presentation characteristics (pre-treatment level of CA 19.9), pancreatic cancer characteristics (location and size), radiological stage (resectable, BR and LA) and TNM stage, neoadjuvant therapy details (regimen, duration, radiotherapy), toxicity defined with CTCAE classification-Common Terminology Criteria for Adverse Events, dropout rate for toxicity/disease progression) and outcomes (OS, PFS and postoperative complications).

Studies were divided into two subgroups according to the comparison reported in the studies; the first comparison was between elderly patients treated with NAT-followed by surgery vs. upfront surgery, the second included young (defined as <70 years) and elderly patients who underwent NAT followed by surgery. A further subgroups analysis was conducted according to the different NAT regimens.

### 2.4. Study Quality Assessment

The methodological quality and potential risk of bias of included observational studies were assessed at outcome level independently by two reviewers using the Newcastle–Ottawa scale [33].

### 2.5. Statistical Analysis

Descriptive statistics will be employed to summarize the characteristics of the included studies. This analysis will include measures such as frequencies, percentages, means, medians, and ranges, as appropriate, to provide an overview of the study populations, interventions, outcomes, and other clinically relevant variables.

## 3. Results

A database search identified 3869 studies. This was reduced to 2815 after the removal of duplicates. A total of 2580 studies were excluded following title review, and a further 204 were removed following assessment of abstracts and papers according to the inclusion criteria, 12 full-text articles met the inclusion criteria (Figure 1), 4 prospective cohort study and 8 retrospective cohort study. The characteristics of the selected study are shown in Table 2.

Among the included studies, seven papers compared NAT and upfront surgery in elderly patients [4,6,24,26,27,28,29] and five compared NAT outcomes among young and elderly patients [5,25,30,31,32].

Characteristics of the Study Populations

Among the 11,385 patients analyzed, 9580 patients (84.1%) were elderly (≥70 years), and, among them, 6359 (66.3%) patients were older than 70 years and 3221 older than 75 years (33.6%). Among the 9580 elderly patients (≥70 years), 5010 (52.2%) were female.

General characteristics of patients, criteria for resectability, and staging are summarized in Table 3.

Elderly populations generally showed poorer performance status, and a higher burden of comorbidities compared to younger patients. ECOG PS was reported in three studies [24,27,31].

Cooper et al. [19] reported a PS ≥ 2 in 48/179 (26.8%) patients; among them, 29 underwent palliative therapy alone.

In Xie et al. [27] ECOG ≥ 2 was reported in the 13.7% of patients ≥ 75 years compared to the 8.2% of younger patients, highlighting the more fragile clinical profile.

The most common cancer location was pancreatic head, neck, or uncinate process in 1595/1945 (82.0%) patients, and pancreatic body and tail in 257/1945 (13.2%) [4,5,24,25,27,28,29,30,31].

In three studies, cancer location was not reported [6,26,32].

Tumor size was only reported in two studies. Barbas et al. [26] reported a mean tumor size of 3.1 ± 1.4 cm, while Rieser et al. [28] reported a median tumor size of 2.45 cm (IQR 2.15–4.0).

In studies evaluating upfront surgery, elderly patients were less likely to receive adjuvant treatments compared to younger patients due to the higher prevalence of perioperative complications or poorer overall health status [4,27]. Moreover, studies comparing young and elderly patients highlighted that elderly patients represent a more vulnerable population, with higher CA19-9 levels and a greater prevalence of advanced disease at diagnosis [25,26].

Among the 9580 elderly patients, only 2307 (24.0%) underwent NAT. NAT regimen and outcomes are reported in Table 4.

Type of NAT regimen was reported in 8/12 studies; FOLFIRINOX and gemcitabine were the most used NAT regimens. Concomitant radiotherapy was reported in 8/12 studies, in 728/1299 patients (56.0%).

The duration of NAT was reported in six studies [5,25,28,29,31,32].

In Qiao et al. [29], FOLFIRINOX was administered with eight bi-weekly cycles for a total of 4 months. In Oba et al. [5], 12 patients were treated with FOLFIRINOX and 22 with GnP (Gemcitabine plus Nab-Paclitaxel) regimens; the median number of cycles was 4 (2–12).

In Weniger et al. [33], FOLFIRINOX or GnP were administered in elderly patients for 5.48 ± 3 cycles.

In studies using S-1 regimens [31,32] hypofractionated external-beam radiotherapy (30 Gy in 10 fractions) with concurrent S-1 (60 mg/m^2^) was administered 5 days per week for 2 weeks prior to pancreatectomy, or with external-beam radiotherapy (50 Gy in 25 fractions) and concurrent S-1 (60 mg/m^2^) for 5 weeks prior to pancreatectomy.

Only two studies reported the CA 19.9 level before and after NAT.

Miura et al. [30] and Suto et al. [32] reported a significant reduction in CA 19.9 levels after NAT, the median value of CA 19.9 was 173 [30] and 213 U/mL [32] before NAT, and 56 [23] and 68 U/mL [24] after NAT.

### 3.1. Comparison of NAT vs. Upfront Surgery in Elderly Patients

Patients from seven studies [4,6,24,26,27,28,29] were compared: 4818 patients who underwent upfront surgery and 2038 patients who underwent NAT followed by surgery.

NAT in elderly patients had a significant positive impact on R0 resection rates compared to upfront surgery. In Nassoiy et al. [4], R0 resection rates were significantly higher in the NAT group than in the upfront surgery group, with clear evidence of tumor downstaging associated with NAT (*p* < 0.001). Similarly, in Rieser et al. [21], elderly patients treated with NAT showed higher R0 resection rates compared to upfront surgery (HR for resectability 0.60, *p* = 0.02).

In Xie et al. [27], elderly patients experienced higher rates of post-operative infections and wound healing difficulties compared to younger (65–75) patients, although no significant differences were observed in severe complications (Clavien-Dindo ≥ 3).

Adjuvant therapy (AT) was mostly administered in the upfront surgery group than the NAT group.

NAT followed by surgery showed an improved overall survival (OS) compared to upfront surgery in elderly patients. The mean OS of the included studies was 27.8 months (15.78–39.1) for NAT patients compared to the 20.3 (11.51–23.8) months of upfront surgery patients.

In Nassoiy et al. [4], patients treated with NAT had a median OS of 27.8 months, significantly higher than the 20.9 months observed in those who underwent upfront surgery (HR = 0.88, *p* < 0.001). Similarly, in Rieser et al. [28], the median OS was 24.6 months in NAT patients versus 17.6 months in those undergoing upfront surgery (HR = 0.60, *p* = 0.02). Data about NAT and upfront surgery in elderly population are summarized in Table 5.

### 3.2. Comparison of NAT in Young vs. Elderly Patients

Patients from six studies were compared [5,25,29,30,31,32], 242 elderly patients and 953 young patients (Table 6).

Elderly patients reported higher NAT adverse events (G3-4) than younger patients, as previously reported.

The R0 resection rates in elderly patients were generally high and comparable to those observed in younger patients, with no significant differences in surgical efficacy. R0 resection was associated with significantly longer survival, regardless of age, confirming that surgical success is a strong predictor of favorable outcomes even in elderly patients [25,29,30].

OS was improved in younger patients than elderly patients in every study included without significant differences.

The mean OS of the included studies was 24.96 months (17.6–27.2) for elderly patients, compared to the 36.2 (23.6–43) months of upfront surgery patients.

In Suto et al., no significant differences in OS were observed between patients ≥ 75 years and younger patients [31,32]. Similarly, in Weniger et al. [25], OS was similar between elderly and younger patients, though elderly patients more frequently received less intensive chemotherapy regimens.

### 3.3. Outcomes of NAT on Resectable, BR and LA Pancreatic Cancer in Elderly

Long-term effects of NAT in elderly patients with different radiological stages (resectable, BR, and LA) were reported in only a few studies.

Rieser ed al [28] compared NAT and upfront surgery in resectable pancreatic cancer reporting a median OS of 24.6 months in NAT patients versus 17.6 months in upfront surgery group (HR = 0.60, *p* = 0.02).

Cooper et al. [24], in their study, evaluated 153 patients with resectable and 23 with BR pancreatic cancer and compared the results of NAT versus upfront surgery. The authors reported a mean OS of 33.8 months in NAT group and 15.1 months in upfront surgery group without differences in resectable and BR groups (*p* = 0.79).

Miura et al. [30] reported a median survival of resectable versus BR patients of 45.0 months versus 31.5 months; the OS was 36.5 months vs. 27.2 months in young (<75 years) and elderly patients (>75 years), respectively.

Weniger et al. [25] reported a mean OS of 23 months and 15 months, respectively, in young (<70 years) and elderly patients (>70 years) without differences according to the stage (BR or LA).

Oba et al. [5] reported on univariate analysis a lower OS in the LA group than BR group (HR:1.72 [1.18, 2.52]) without differences of NAT according to age impacting this analysis.

### 3.4. Secondary Endpoints Analyses (Resection Rate, Toxicity, Dropout, NAT Regimens Analysis, TTR)

The number of elderly patients who reached surgery after NAT varies among the different studies. In only three studies, the resection rate was not reported [4,6,29]. The resection rate varied from 48% to 100% (Table 4).

NAT major toxicity was reported in only four studies [29,30,31,32], ranging from 17% to 57.5%.

Patients’ dropout from curative treatment due to NAT toxicity and/or disease progression was reported in four studies [5,24,25,31], ranging from 21.2% to 44.0%.

Cooper et al. [24] reported that neoadjuvant treatment consisted of external-beam radiation (30 Gy in 10 fractions, or 50.4 Gy in 28 fractions) with concurrent gemcitabine, 5-FU, or capecitabine. In contrast, Rieser et al. [28] described a neoadjuvant regimen primarily based on gemcitabine, with a median of three cycles administered in 94% of cases. Only 2% of patients received 5-fluorouracil-based therapy alone, while 4% received a combination of both gemcitabine- and 5-fluorouracil-based regimens. The study reported a similar R1 resection rate between the NAT and upfront surgery groups.

Qiao et al. [29] analyzed 40 elderly patients treated with FOLFIRINOX. Only 65% of elderly patients completed the intended neoadjuvant FOLFIRINOX compared to 81.4% of younger patients. Elderly patients experienced higher rates of major toxicities (grades 3 and 4) than younger patients (57.5% vs. 40.2%); however, this did not impact surgical resection rates, which appeared improved in the subgroup treated with neoadjuvant therapy (68.5% vs. 47.6%).

Miura et al. [30] reported that neoadjuvant treatment consisted of gemcitabine-based chemoradiation for resectable patients and chemotherapy followed by chemoradiation for borderline resectable (BR) patients. Both gemcitabine-based NAT and FOLFIRINOX were used in elderly patients. Major adverse events or disease progression occurred in 31% of elderly patients compared to 25% of younger patients. The authors suggested that personalized treatment regimens could reduce the burden of adverse events while maintaining treatment efficacy. The dropout rate was 33%.

Weniger et al. [25] compared the standard FOLFIRINOX regimen with the standard GnP regimen, highlighting higher rates of hematological toxicity in elderly patients, particularly neutropenia and anemia, especially with intensive regimens such as FOLFIRINOX. Less intensive regimens were better tolerated in elderly patients, allowing a higher percentage to complete treatment.

Similarly, Oba et al. [5] compared FOLFIRINOX with the GnP regimen and reported worse outcomes for the latter.

Suto et al. [31,32] reported increased rates of hematological and gastrointestinal toxicities in patients aged ≥ 75 years compared to younger patients. However, these toxicities rarely prevented patients from completing treatment or undergoing surgery.

TTR has been reported in several studies, along with other additional outcomes. In Weniger et al. [25], TTR was shorter in elderly patients compared to younger ones (*p* = 0.019), this did not negatively impact OS. Suto et al. [31,32]. reported similar rates between young and elderly patients, demonstrating that NAT effectively prevents disease recurrence, even in older patients.

In Qiao et al. [29], TTR was reported to be slightly higher in the NAT group (8.8 months) than in the upfront surgery group (7.1 months).

## 4. Discussion

NAT is an effective therapeutic strategy for elderly patients with resectable or BR or LA pancreatic cancer, improving overall survival and R0 resection rates compared to upfront surgery. Although elderly patients have a higher risk of toxicity, the use of personalized chemotherapy regimens minimizes adverse events and ensures safe access to surgery.

Although the median OS for elderly patients treated with NAT is lower compared to younger patients (24.96 months vs. 36.2 months), the data highlight that, when feasible, a neoadjuvant strategy in elderly patients still provides a statistically significant survival benefit compared to upfront surgery (27.8 months [15.78–39.1] vs. 20.3 months [11.51–23.8], *p* < 0.0001).

In contrast with the current literature data [13], despite the limitations of the analyzed study designs (observational), a benefit was also observed in patients with resectable disease, which is maintained in elderly patients. Conversely, LA pancreatic cancer confirms poorer survival outcomes compared to earlier stages of the disease.

Weniger et al. found that NAT in elderly patients had similar benefits than younger patients, even with a higher incidence of hematologic and gastrointestinal adverse events [25]. These results also confirm the role of NAT in elderly patients in which age is not a mere factor that excludes patients from curative treatments [30,31]. These findings are supported in metastatic settings, where the combination of chemotherapy as shown significantly better outcomes than monotherapy; regimens like GnP outperform monotherapy in terms of disease control and overall survival [34].

The tolerability of NAT in elderly patients is a critical challenge. Adverse events such as neutropenia, anemia, and gastrointestinal toxicity occur more frequently in elderly patients compared to younger ones [30,31]. Less intensive regimens like GnP have shown superior tolerability compared to aggressive protocols like FOLFIRINOX, enabling more elderly patients to complete treatment [25]. The different toxicity profiles of the two regimens are widely confirmed by real-world data, with greater hematological toxicity, particularly febrile neutropenia, associated with the FOLFIRINOX regimen, and better overall tolerability of the GnP regimen, though it is linked to a higher incidence of peripheral neuropathy [35]. This factor is crucial in selecting the most appropriate regimen for more fragile patients. Miura et al. addressed that personalized treatment regimens can reduce toxicity burden while maintaining efficacy [30].

The therapy dropout rate in elderly patients ranges from 21.2% to 44%, primarily due to toxicity or disease progression. This underscores the need for careful patient selection based on performance status (ECOG) and comorbidities rather than age alone [23,26]. Employing geriatric assessments could have a strong rationale for identifying ideal candidates for NAT. Tools like frailty assessments and toxicity risk scores have been explored in pancreatic cancer management but remain underdeveloped in NAT-specific settings. Studies highlighted the need for developing robust geriatric assessment tools, as this represents a challenging area of research [36].

The dataset presents significant limitations that preclude the feasibility of conducting a robust meta-analysis. While the systematic review identified studies addressing the role of NAT in elderly patients with pancreatic cancer, the majority of these studies lack critical data necessary for meta-analytic synthesis. Specifically, key parameters such as hazard ratios (HRs) for OS, confidence intervals (CIs), and event counts are either absent or incomplete in most studies. Without these parameters, it was not possible to accurately calculate pooled effect sizes or assess the consistency of results across studies. Moreover, the studies included in the dataset exhibit substantial clinical and methodological heterogeneity. A major source of heterogeneity is the inconsistent definition of “elderly” across studies. While some studies define elderly patients as those aged ≥ 70 years, others use cutoffs such as ≥75 years. This variability introduces differences in baseline patient characteristics, comorbidities, and treatment tolerability, which can influence outcomes.

The heterogeneity of treatment modalities used precluded meaningful analysis of the impact of specific NAT regimens.

Another critical limitation lies in the retrospective nature of all included studies. Retrospective designs are inherently prone to selection bias, variability in treatment protocols, and differences in follow-up practices. Pooling results from such studies without adequate control of these biases could amplify these limitations rather than mitigate them. The findings highlight the need for prospective studies or RCTs with standardized definitions and consistent reporting of key outcomes. This is particularly important given that real-world data reflect a substantial population of elderly patients with advanced pancreatic cancer—a group that is often underrepresented in clinical trials. In many RCTs, the median age of enrolled patients tends to be significantly lower than that of the real-life population, and age-related exclusion criteria frequently limit the inclusion of older patients. This discrepancy poses substantial challenges to the generalizability of trial results, as they may not accurately reflect the outcomes or tolerability of therapies in the elderly population most affected by the disease. Addressing these gaps through better-designed studies would provide more reliable evidence to guide treatment decisions in this often-overlooked demographic.

## 5. Conclusions

NAT is associated with improved survival and surgical outcomes in elderly pancreatic cancer patients, despite a higher risk of adverse events. Patient selection based on performance status rather than age alone is essential to optimize treatment benefits. Further prospective trials are needed to refine treatment approaches in this population.

## Figures and Tables

**Figure 1 cancers-17-00747-f001:**
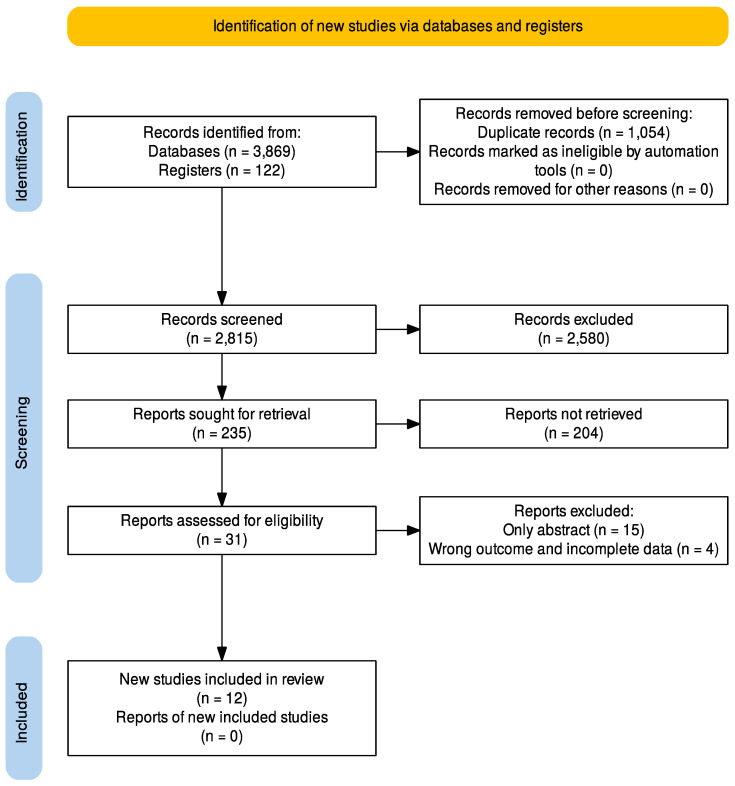
PRISMA 2020 flowchart.

**Table 1 cancers-17-00747-t001:** PICO Framework and the Question Statement.

Criteria	PICOS Question
Population	Elderly patients (>70 years) with pancreatic cancer
Intervention	Neoadjuvant chemotherapy followed by surgery
Comparison	Neoadjuvant chemotherapy vs. upfront surgery; neoadjuvant chemotherapy in young (<70 years) and elderly patients (>70 years)
Outcomes	OS, TTR, surgical resection rate, major toxicity
Study	RCT, cohort studies, case control studies, case series

Legend: OS: Overall survival; TTR: time-to-recurrence, RCT: Randomized controlled trial.

**Table 2 cancers-17-00747-t002:** Characteristics of included studies.

Reference	Study Design	Patients n	Study Population	Comparison	Quality Score
Barbas 2012 [26]	RCS	203	≥75 yeas	NAT vs. upfront surgery	6
Cooper 2014 [24]	RCS	179	>70 years	NAT vs. upfront surgery	7
Xie 2019 [27]	PCS	903	>75 years	NAT vs. upfront surgery	8
Groen 2020 [6]	PCS	3624	≥70 years	NAT vs. upfront surgery	6
Rieser 2021 [28]	RCS	158	≥75 years	NAT vs. upfront surgery	7
Qiao 2023 [29]	RCS	341	≥75 years	NAT vs. upfront surgery NAT in <75 vs. NAT in ≥75	8
Nassoiy 2023 [4]	RCS	5086	≥70 years	NAT vs. upfront surgery	7
Miura 2015 [30]	PCS	246	<75 and ≥75	NAT in <75 vs. NAT in ≥75	8
Weniger 2020 [25]	RCS	165	≤70 and >70	NAT in ≤70 vs. NAT in >70	8
Oba 2021 [5]	PCS	246	<70; 70–74; ≥75	NAT in <70 vs. 70–74 vs. ≥75	8
Suto 2023 [31]	RCS	122	<75 and ≥75	NAT in <75 vs. NAT in ≥75	7
Suto 2024 [32]	RCS	112	<75 and ≥75	NAT in <75 vs. NAT in ≥75	7

Legend: PCS: prospective cohort study; RCS: retrospective cohort study; NAT: Neoadjuvant chemotherapy.

**Table 3 cancers-17-00747-t003:** Elderly patients’ general characteristics.

Reference	Elderly Patients (n)	Gender F (n)	Elderly NAT (n)	R (n)	BR (n)	LA (n)	Stage I (n)	Stage II (n)	Stage III (n)
Barbas 2012 [26]	32	19	22	NR	NR	NR	NR	NR	NR
Cooper 2014 [24]	179	85	153	153	26	0	NR	NR	NR
Xie 2019 [27]	384	177	83	NR	NR	NR	88	296	0
Groen 2020 [6]	2563	1977	800	NR	NR	NR	1105	2519	0
Rieser 2021 [28]	158	98	68	158	0	0	39 *	60 *	56 *
Qiao 2023 [29]	138	24	52	NR	NR	NR	31	22	1
Nassoiy 2023 [4]	5086	2640	939	NR	NR	NR	431	508	0
Miura 2015 [30]	36	15	36	19	17	0	9 *	14 *	0
Weniger 2020 [25]	33	17	33	0	19	14	NR	NR	NR
Oba 2021 [5]	34	18	34	0	24	10	NR	NR	NR
Suto 2023 [31]	44	23	44	33	11	0	1 *	36 *	2 *
Suto 2024 [32]	43	21	43	31	12	0	21 *	16 *	NR

Legend: NAT: Neoadjuvant chemotherapy; R: Resectable; BR; Borderline Resectable; LA: Locally Advanced; NR: Not Reported. * Stage was defined after NAT + surgery.

**Table 4 cancers-17-00747-t004:** NAT outcomes.

Reference	NAT n	CA 19.9 Level Before NAT (IQR)	NAT Regimen n	RT n (%)	Major Toxicity > 3 N (%)	Dropout for NAT Toxicity/Disease Progression (%)	CA 19.9 Level After NAT (IQR)	Surgery N (%)
Barbas 2012 [26]	22	NR	NR	NR	NR	NR	NR	22 (100%)
Cooper 2014 [24]	153	157 (70–520)	Gemcitabine ± cisplatin or erlotinib	NR	NR	37 (24.1%)	NR	74 (48%)
Xie 2019 [27]	83	210 (54–756)	NR	46 (55.4%)	NR	NR	NR	83 (100%)
Groen 2020 [6]	800	NR	NR	NR	NR	NR	NR	NR
Rieser 2021 [28]	68	142 (46–472)	FOLFIRINOX OR gemcitabine/ abraxane	10 (5.0%)	NR	NR	NR	68 (100%)
Qiao 2023 [29]	52	23 patients (42.6%) ≥ 37 U/mL	FOLFIRINOX	41 (75.9%)	30 (57.5%)	NR	NR	NR
Nassoiy 2023 [4]	939	484 patients (51.5) ≥ 38 U/mL	NR	481 (51.2%)	NR	NR	NR	NR
Miura 2015 [30]	36	173 (296)	Gemcitabine 5-FU–based, FOLFIRINOX	34 (94.7%)	6 (17.0%)	12 (33.3%)	56 (691)	24 (66.6%)
Weniger 2020 [25]	33	NR	FOLFIRINOX Gemcitabine/ nab-paclitaxel	NR	NR	7 (21.2%)	NR	28 (84.8%)
Oba 2021 [5]	34	126 (1–6490) *	FOLFIRINOX Gemcitabine/ nab-paclitaxel	29 (85.2%)	NR	15 (44.0%)	NR	19 (56.0%)
Suto 2023 [31]	44	NR	S	44 (100%)	10 (23.0%)	NR	NR	39 (89.0%)
Suto 2024 [32]	43	213 (19–813)	S-1, Gemcitabine	43 (100%)	10 (23.0%)	NR	68 (10–209)	38 (88.0%)

Legend: NAT: Neoadjuvant chemotherapy; RT: Radiotherapy; NR: Not Reported. * Range.

**Table 5 cancers-17-00747-t005:** Comparison of NAT vs. upfront surgery in elderly patients.

Reference	NAT + S n	US n	Margin Positive (R 1–2) After NAT n (%)	Margin Positive (R 1–2) After US n (%)	AT in NAT Group n (%)	AT in US Group n (%)	OS NAT + S Months	OS US Months
Barbas 2012 [26]	22	10	NR	NR	9 *		23.7 *	
Cooper 2014 [24]	74	26	NR	NR	37 (24.1%)	11 (48%)	33.8	15.1 (5.4–100.8)
Xie 2019 [27]	83 **	195	NR	NR	121 *		39.1	23.8
Groen 2020 [6]	800 **	266	NR	NR	NR	NR	24	22
Rieser 2021 [28]	68	90	24 (34.0%)	26 (29%)	34 (51%)	43 (48%)	24.6	17.6
Qiao 2023 [29]	52 **	84	17 (31.5%)	44 (52.4%)	NR	NR	15.78	11.51
Nassoiy 2023 [4]	939 **	4147	159 (16.9%)	934 (22.5%)	NR	2627 (63.3%)	27.8	20.9

Legend: NAT: Neoadjuvant chemotherapy; NAT + S: Neoadjuvant chemotherapy + surgical resection; US: Upfront Surgery; AT: Adjuvant chemotherapy; OS: Overall Survival; NR: Not Reported. * The number (%) refers to both groups (NAT + S and US), ** The number refers to each patient treated with NAT and not always followed by surgical resection in both groups.

**Table 6 cancers-17-00747-t006:** Comparison of NAT in young and elderly patients.

Reference	Elderly * n	Young ** n	Margin Positive (R 1–2) Elderly n (%)	Margin Positive (R 1–2) Young n (%)	OS Elderly Months	OS Young Months
Qiao 2023 [29]	52	273	17 (31.5)	50 (17.4)	16.43	30.83
Miura 2015 [30]	36	210	2 (8.0%)	1 (1.0%)	27.2	36.5
Weniger 2020 [25]	33	132	14 (53.9%)	51 (50.0%)	26.0	35.0
Oba 2021 [5]	34	191	NR	NR	17.6	23.6; 18.0 *
Suto 2023 [31]	44	78	3 (8.0%)	0 (0%)	27	43.0
Suto 2024 [32]	43	69	4 (11.0%)	0 (0%)	27	43.0

Legend: OS: Overall Survival; NR: Not Reported. * Young and elderly were reported according to the differences found by each study. The cut-off to define elderly population is ≥75. Weniger et al. [25] defined elderly with the cut-off age of >70 year. Oba et al. [5] divided patients into 3 groups according to age, <70, 70–74 and ≥75. ** OS in patients <70 was 23.6 months and, in patients between 70 and 74 years, OS was 18.0 months.

## Data Availability

The raw data supporting the conclusions of this article will be made available by the authors on request.

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
