# Peer review of "Neoadjuvant Treatment in Localized Pancreatic Cancer of the Elderly: A Systematic Review of the Current Literature"

_cancers, 2025, doi:10.3390/cancers17050747_

Round 1

Reviewer 1 Report

Comments and Suggestions for Authors

Dear Authors, to write a sound review on NAT in pancreatic cancer more data are needed.
First of all, the site of the neoplasm: head, body, hooked process or tail. Then consider the pre-NAT staging in other words what involvement of the vessels? And how many patients received biliary stents? And radiotherapy at least defines how many sessions or at least  location So I would recommend rewriting the article, which in any case has points of interest, and adding more details. My advice to the authors is: analyze a low number of article published but upper level and containing more useful details.
As it currently stands, it cannot be published in a high-IF journal

Author Response

Dear Reviewer many thanks for your suggestions.

We improved the introduction. We expanded methods sections including more details like site of neoplasm, CA 19.9 levels, number of NAT cycles and RT details when reported. I’m sorry but other details cannot be included (biliary stent, details of radiotherapy such as duration or site) because they have not been reported. We reported every adding information reported in the studies.

Only few studies evaluated the effect of NAT on elderly patients. This is one of the first review on this topic.

The main goal of this study was to evaluate the effect of NAT on older population including short-term results like toxicity and long-term results including NAT impact on OS.

We believe that this study could help the clinicians to consider NAT also in elderly people who could really benefit from this therapy.

Reviewer 2 Report

Comments and Suggestions for Authors

The authors conducted the systematic review of the literatures for neoadjuvant therapy 9NAT) for elderly patients with pancreatic cancer (PDAC). The topic is important in this field and the study design appeared sound. The manuscript was overall well-written. My major critique for this manuscript was the lack of analysis for the impact of NAT on the long-term outcomes for patients with resectable PDAC vs borderline resectable (BR)/locally advanced unresectable (LAUR) PDAC. Given that the indication of NAT for resectable PDAC remains controversial in contrast to the one for those with BR/LAUR PDAC, over-simplified message for advantage of NAT for elderly patients was misleading.

Author Response

Dear Reviewer many thanks for your suggestions.

On the first manuscript draft we were undecided about the inclusion of different OS for resectable, BR or LA pancreatic cancer due to the limited available data.

We understand and share your thoughts on this topic.

To overcome misleading messages, we rewrote the results section adding a proper section reporting when possible the different results in term of OS among resectable, BR and LA.

Despite the limited data they could make easier to understand the paper.

Reviewer 3 Report

Comments and Suggestions for Authors

Peer Review Report

Title: Neoadjuvant treatment in localized pancreatic cancer of the elderly: a systematic review of the current literature

Authors: Elena Orlandi et al.

1. Introduction. The introduction provides a solid background on pancreatic cancer and the rationale for neoadjuvant therapy (NAT) in elderly patients. However, certain references could be expanded to include more recent clinical trials or meta-analyses. Additionally, the discussion on the lack of consensus regarding NAT in elderly patients could benefit from a broader comparison with similar malignancies where NAT is more established.

2. Research Design. The systematic review adheres to PRISMA guidelines, and the use of the Newcastle-Ottawa scale for quality assessment is appropriate. However, there is a notable heterogeneity among included studies, especially concerning the definition of "elderly" and treatment regimens. A meta-analysis component would have strengthened the findings.

3. Methods. The methodology is well-described, including the search strategy, inclusion/exclusion criteria, and data extraction. However, while the authors acknowledge selection bias, a more detailed discussion on how heterogeneity was addressed would improve clarity. Moreover, defining how toxicity grades were assessed across different studies would enhance reproducibility.

4. Results. The results are clearly presented, with well-structured tables summarizing the key findings. The distinction between NAT versus upfront surgery and young versus elderly subgroups is well-articulated. A more in-depth discussion of specific chemotherapy regimens and their relative tolerability would be beneficial.

5. Conclusions The conclusions are consistent with the results and emphasize the importance of performance status over chronological age in treatment decision-making. The authors appropriately call for prospective trials to validate their findings.

  • Originality/Novelty: The topic is relevant, but there are other reviews on similar themes; more emphasis on novel aspects would strengthen impact.
  • Significance of Content: The study addresses a clinically significant gap in the literature.
  • Quality of Presentation: Well-structured, but minor clarifications needed in methods and discussion.
  • Scientific Soundness: Appropriate methodology and statistical analysis, though meta-analysis would be preferable.
  • Interest to Readers: Relevant to oncologists, surgeons, and researchers in pancreatic cancer.
  • Overall Merit: Strong contribution to literature, with minor refinements suggesComments

Comments and Suggestions for Authors

  1. Clarify heterogeneity handling: The study acknowledges variability in defining "elderly." A more explicit discussion of how this was addressed statistically or narratively would be beneficial.
  2. Expand chemotherapy discussion: More details on the impact of specific NAT regimens on toxicity and outcomes would enhance clinical applicability.
  3. Strengthen introduction: Including more recent trials and systematic reviews comparing NAT versus upfront surgery in pancreatic cancer would provide better context.
  4. Consider a meta-analysis approach: If feasible, a meta-analysis would provide more robust conclusions regarding survival benefits and toxicities.

Author Response

Dear Reviewer many thanks for your suggestions.

We updated the manuscript according to your suggestions.

We try to make the paper easier to understand. We modified the text to reduce the risk of confusion about elderly definition, we reported that we considered elderly only patients > 70 years including in the results data on patients older of 70 years, we reported that some studies reported data of elderly patients older than 75 years.

We discussed about more details on specific NAT regimens

Introduction. We add some references including more recent clinical trials.

Research Design

The heterogeneity of the included studies was assessed considered as elderly patients older than 70 years.

In the included studies elderly cut-off age vary from 70 to 75 years. Only 1 study reported in the text 65 years old as cut-off but after the definition the authors stratified the patients in 65-74 and older than 75 years. In the end we considered elderly patients with 70 years old.

The impact of different NAT regimens was updated in this version adding a specific section in the results.

A meta-analysis component was considered in the original study idea but after complete paper’s selection the data were not adequate for meta-analysis. We talked with 2 statisticians to evaluate the correct methodology and both disagreed about the idea of meta-analysis.

We specified the impossibility to perform meta-analysis on discussion in the limitation paragraph

We hope that you understand the reasons about our choice.

Methods

We expanded the methods including a detailed discussion on how heterogeneity was addressed.

Toxicity was reported according to the general classification of CTCAE based on the reports in the different studies when reported.

Results

Accordingly with your suggestions we add a more in-depth discussion of specific chemotherapy regimens reporting the impact and outcomes of different NAT regimens.

Round 2

Reviewer 1 Report

Comments and Suggestions for Authors

The Paper  has improved and appears more scientifically sound. Even if the topic has been little studied (relatively) you must remember to always select the articles you want to propose because the weight as IF is often very different. In addition, it is necessary to clarify what you want to highlight and for this it is necessary to classify patients with scientific precision. Even just the site of the tumor is useful. I would advise you to expand your next study on this topic by reporting especially the studies where there is definition of vascular invasion and surgical reporting. You will need more time for genetic study, but that's another story

Author Response

Dear Reviewer,

Many thanks for your work and your suggestions. Obviously the paper lacks of several details that cannot be addressed at this moment. We hope that future well designed papers could provide more details.

Thank you again for your comments.

Reviewer 2 Report

Comments and Suggestions for Authors

The authors revised their manuscript according to the reviewer's comments, and the manuscript has improved. It warrants for publication.

Author Response

Dear Reviewer,

Thank you for your comments.